# Interstitial cystitis/bladder pain syndrome patient is associated with subsequent increased risks of outpatient visits and hospitalizations: A population-based study

Kun-Lin Hsieh[1,2], Hung-Yen Chin[3], Tsia-Shu Lo[4,5], Cheng-Yu Long[6,7], Chung-Han Ho[8], Steven Kuan-Hua Huang[1], Yao-Chi Chuang[9], Ming-Ping Wu[10,11]*

1 Division of Urology, Department of Surgery, Chi Mei Medical Center, Tainan, Taiwan, 2 Department of Environmental and Occupational Health, College of Medicine, National Cheng Kung University, Tainan, Taiwan, 3 Department of Obstetrics and Gynecology, Taipei Medical University Hospital, School of Medicine, College of Medicine, Taipei Medical University, Taipei, Taiwan, 4 Division of Urogynecology, Department of Obstetrics and Gynecology, Linkou, Chang Gung Memorial Hospital, Linkou Medical Center, Taoyuan, Taiwan, Republic of China, 5 School of Medicine, Chang Gung University, Taoyuan, Taiwan, Republic of China, 6 Department of Obstetrics and Gynecology, Kaohsiung Medical University Hospital, Kaohsiung Medical University, Kaohsiung, Taiwan, 7 Department of Obstetrics and Gynecology, Siaogang Hospital, Kaohsiung Medical University, Kaohsiung, Taiwan, 8 Department of Medical Research, Chi Mei Medical Center, Tainan, Taiwan, 9 Department of Urology, Kaohsiung Chang Gung Memorial Hospital, Chang Gung University College of Medicine, Kaohsiung, Taiwan, 10 Division of Urogynecology, Department of Obstetrics and Gynecology, Chi Mei Medical Center, Tainan, Taiwan, 11 Department of Obstetrics and Gynecology, College of Medicine, Fu-Jen Catholic University, Taipei, Taiwan

* mpwu@mail.chimei.org.tw

**Data Availability Statement:** All relevant data are within the manuscript.

## Abstract

Interstitial cystitis/bladder pain syndrome (IC/BPS) is not only a chronic urinary bladder pain syndrome but is also associated with multifactorial etiology. Our study aimed to test the hypothesis that IC/BPS is associated with subsequent increased risks of outpatient visits and hospitalizations. Using nationwide database, the diagnoses were based on the International Classification Codes (ICD-9-CM) (595.1) of at least three outpatient services during 2002–2008, (n = 27,990) and cystoscopic finding Hunner type and/or glomerulation with pre-audit criteria. All recruited cases monitored for subsequent outpatient visits and hospitalizations for 2 years, including all-cause and specialty-specific departments, were classified according to medical specialty and age group (<40, 40–60, ≥60 years of age). IC/BPS patients have more overall outpatient department (OPD) visits and an overall adjusted incidence rate ratio (IRR) of 1.64. As for specialty, IRRs were higher in psychiatry (2.75), Chinese medicine (2.01), and emergency medicine (2.00), besides urology and gynecology. The IRRs decreased as age advanced (2.01, 1.71, and 1.44, respectively), except for gynecology (2.42, 2.52, and 2.81). A similar phenomenon happens in hospitalization with IRR of 1.69. Due to claim data characteristics, whether ulcer type IC/BPS findings can be deductive to non-ulcer type remains inclusive. Current results indicate the impacts of healthcare burden in broad spectrum about IC/PBS patients. IC/BPS has been suggested to be associated with lower threshold of healthcare visits and some coexisting disease and is comprised of systemic dysregulation, and is beyond the scope of local bladder-urethra disease. Adequate

**Funding:** The author(s) received no specific funding for this work.

**Competing interests:** The authors have declared that no competing interests exist.

recognition of associated or comorbid factors and possible recommendation or referral for IC/BPS patients can help provide better healthcare quality.

## Introduction

Interstitial cystitis/bladder pain syndrome (IC/BPS) is a chronic pain syndrome defined by an unpleasant sensation (pain, pressure, discomfort) perceived to be related to the urinary bladder in conjunction with lower urinary tract symptoms (LUTS), including urgency, frequency, and nocturia, of more than 6 weeks' duration [1]. IC/BPS is associated with a wide variety of clinical phenotypes with diverse etiologies [2]. Some candidate etiologies of IC/BPS have been studied and reported, including inflammatory processes or infection, autoimmunity, mast cell activation, local neuronal dysfunction or mucosal abnormalities, and cross-organ sensitization between visceral organs [3]. However, the real etiology is still inconclusive. IC/BPS was recognized as a local disease before, and treatment was mainly focused on focal symptomatic management. Thereafter, multifactor effects are expounded by most studies.

In addition to the bladder/urethra, IC/BPS is often associated with multiple organ dysfunction. Current hypothesis suggests that IC/BPS is a systemic disorder or a member of a family of hypersensitivity disorders that affects the bladder and other somatic/visceral organs [4]. Keller et al. reported that subjects with IC/BPS had an increased prevalence of multiple comorbidities, such as cardiovascular, neurologic, rheumatological, pulmonary, endocrine, renal gastrointestinal, viral/infectious, hematological, mental illness, oncological, and other diseases [5]. Watkins et al. found that probable current depression and panic attack rates are high, associated with IC/BPS patients with poor quality of life, in a community-based sample [6]. McKernan et al. indicated severe psychological impairment among IC/BPS patients, including anxiety and depression, in a systematic review [7]. Therefore, IC/BPS is possibly associated with subsequent increased risk of medical service, including OPD visits and hospitalizations. Furthermore, higher risk of medical service will lead to larger economic burden to the healthcare system, even though the prevalence of IC/BPS is not so high in the general population [8].

This study hypothesizes that IC/BPS is associated with subsequent increased risk of OPD visits and hospitalizations. Based on a nationwide population-based database among National Health Insurance (NHI) enrollees in Taiwan, this study examines whether IC/BPS is a risk factor of other medical or surgical conditions and the age effect on the healthcare impacts.

## Materials and methods

### Data source

This study was based on data from National Health Insurance Research Database (NHIRD) in Taiwan. The detailed information of NHIRD was described in our previous report [9]. Briefly, the NHI system was implemented from 1995 and has covered almost all population (98%) in Taiwan by the compulsory and universal health insurance. The diagnostic codes used in the NHIRD are based on the International Classification of Diseases, Ninth Revision, Clinical Modification (ICD-9 CM) codes. Except for the ICD-9 CM, the NHIRD provides information of encrypted patient identification numbers, sex, date of birth, dates of OPD visits, inpatient expenditures by hospitalizations, etc. The study was conducted according to the regulations of the NHIRD and obtained exemption from approval by the Institutional Review Board of the Chi Mei Medical Center (IRB10310-E01).

## Assembly of study participants

Individuals included in the study group (IC/BPS group, n = 27,900) were identified as those who had at least three outpatient service claims during 2002–2008 with the ICD-9 CM code 595.1. To restrict our study group in more well-defined subjects, only IC/BPS patients who received bladder instillation of sodium hyaluronate were recruited. According to Taiwan's NHI regulation, the prescription of sodium hyaluronate (Cystistat; Bladder Healthy UK, Birmingham, UK) requires strict pre-audit criteria. Its provision included the presence of glomerulation or Hunner's lesions under cystoscopy after hydrodistention test, voiding diary, negative urine cytology, and urine analysis. The same methods for the enrollment of IC/BPS patients have already been applied in a previous study of ours and other published studies [9,10].

The control group (non-IC/BPS group matched one-to-one to each IC/BPS patient, n = 27,990) was composed of patients not diagnosed with IC/BPS and randomly selected from the dataset. They were matched by sex, age, index date, and comorbidities, including hypertension (HTN; ICD-9-CM: 401–405), diabetes mellitus (DM; ICD-9-CM: 250), hyperlipidemia (ICD-9-CM: 272), and coronary artery disease (CAD; ICD-9-CM: 410–414). The index date for participants with IC/BPS was the date of their first registration. After the index date, patients would receive survey, assessment, and treatment of sodium hyaluronate instillation. Thus, patients who had outpatient visits with IC/BPS code at least three times after index date into study group have been selected.

All recruited cases of IC/BPS and the non-IC/BPS control group were monitored for subsequent OPD visits and hospitalizations for 2 years after the index date or until the expired day in the follow-up period. Subsequent OPD visits and hospitalizations of healthcare services, including all-cause and specialty-specific OPD visits and hospitalizations, were recorded. The medical services were classified according to medical specialty and age group (<40, 40–60, ≥60 years of age).

## Measures and statistical analysis

Demographical information of the patients was obtained from the NHI enrollees, including age, gender, insurance amount, and region. Age was grouped into the following three categories: 18–39, 40–59, and 60 or more years of age. The insurance amounts of the enrollees were classified as follows: less than US$640 (NTD20,000), US$640–US$1280 (NTD 20,000–39,999), and more than US$1,281 (NTD 40,000). The geographic distributions were classified into northern, central, southern, and eastern.

McNemar's test for paired categorical variables was conducted to compare differences between the IC/BPS and non-IC/BPS groups regarding sociodemographic characteristics and comorbidities. The incidence rate (IR) was calculated as the times of medical services during the follow-up period, divided by the total person years for each group. The subsequent risk of OPD visits and hospitalization were estimated using Poisson regression for the incidence rate ratio (IRR). Poisson regression model was used to estimate the adjusted IRR between the IC/BPS and non-IC/BPS groups using the SAS procedure, after considering the possible overdispersion issue and potential confounders, including age, gender, income, area, HTN, DM, hyperlipidemia, and CAD. Some other potential confounders were not available due to claim data characteristics, including education, marital status, alcohol use, tobacco use, obesity, etc. Subspecialties were categorized under internal medicine or surgery, unless otherwise specified (e.g., neurology and neurosurgery). All analyses were performed using SAS software version 9.4 (SAS Institute, Cary, NC, USA). For the descriptive statistical analysis, a p-value of less than 0.05 was considered significant, and a more precise p-value of 0.0025 (0.05/20) with

Bonferroni correction (significance level/number of specialty departments) was considered significant because of the multiple hypothesis tests performed.

## Results

A total of 27,900 patients were included into each IC/BPS and control group. The participants were matched for age, gender, and underlying disease (DM, HTN, and CAD) in the two groups. The demographic information for IC/BPS and non-IC/BPS individuals is shown in Table 1.

OPD visits and hospitalizations of the IC/BPS and control groups were identified. The units of the visits or hospitalizations are expressed as number per one person-year. As expected, patients with IC/BPS had higher OPD visits in gynecology (3.57 for the IC group vs. 1.42 for the control group) (number per one person-year) and urology (1.24 vs. 0.19), with an adjusted IRR of 2.51 (95% confidence interval [CI], 2.43–2.60) and 6.68 (95% CI, 6.22–7.17), respectively. Gender difference was observed in urology visits among IC/BPS patients with an IRR of 1.07 (females) and 2.33 (males), and the adjusted IRR (females to males) was 0.42 (95% CI, 0.40–0.45). The total OPD visit rate (except for gynecology and urology) (number per one person-year) for the IC/BPS and non-IC/BPS groups was 27.74 and 16.96, respectively. After adjusting for age, gender, HTN, DM, hyperlipidemia, and CAD, the adjusted IRR was 1.64 (95% CI, 1.62–1.66). In addition to the IRRs for gynecology and urology, those for other departments were significantly higher as well as in psychiatry (2.75; 95% CI, 2.50–3.01), Chinese medicine (2.01; 95% CI, 1.94–2.08), emergency room (2.00; 95% CI, 1.89–2.11), and others. All p-values were <0.001. Data are shown in detail in Table 2.

**Table 1. Demographic information about interstitial cystitis/bladder pain syndrome (IC/PBS) patients and non-IC/PBS controls.**

|  | IC/BPS (N = 27,990) | Non-IC/BPS (N = 27,990) | p-value* |
|---|---|---|---|
| Age |  |  |  |
| <40 | 8,633 (30.84) | 8,633 (30.84) | >0.9999 |
| 40–60 | 11,689 (41.76) | 11,689 (41.76) |  |
| >60 | 7,668 (27.40) | 7,668 (27.40) |  |
| Gender Female | 22,042 (78.75) | 22,042 (78.75) | >0.9999 |
| Male | 5,948 (21.25) | 5,948 (21.25) |  |
| Area Northern | 15,572 (55.63) | 14,321 (51.16) | <0.0001 |
| Central | 4,370 (15.61) | 4,932 (17.62) |  |
| Southern | 7,391 (26.41) | 8,116 (29.00) |  |
| Eastern | 657 (2.35) | 621 (2.22) |  |
| Income <20,000 | 13,602 (48.60) | 14,263 (50.96) | <0.0001 |
| 20,000~40,000 | 10,906 (38.96) | 10,249 (36.62) |  |
| >40,000 | 3,482 (12.44) | 3,478 (12.43) |  |
| Comorbidity |  |  |  |
| HTN Yes | 5,492 (19.62) | 5,492 (19.62) | 1.0000 |
| No | 22,498 (80.38) | 22,498 (80.38) |  |
| DM Yes | 2,379 (8.50) | 2,379 (8.50) | 1.0000 |
| No | 25,611 (91.50) | 25,611 (91.50) |  |
| Hyperlipidemia Yes | 2,146 (7.67) | 2,146 (7.67) | 1.0000 |
| No | 25,844 (92.33) | 25,844 (92.33) |  |
| CAD Yes | 1,953 (6.98) | 1,953 (6.98) | 1.0000 |
| No | 26,037 (93.02) | 26,037 (93.02) |  |

*p-value is from McNemar's test for matched pairs.

**Table 2. Incidence rate ratio (IRR) of outpatient visits between IC/BPS and non-IC/BPS individuals.**

| Department | IC/BPS (N = 27990) | | Non-IC/BPS (N = 27990) | | Unadjusted | | Adjusted | |
|---|---|---|---|---|---|---|---|---|
| | Visits | Rate* | Visits | Rate* | IRR** (95% CI) | p-value | IRR***(95% CI) | p-value |
| Gynecology# | 156665 | 3.57 | 62354 | 1.42 | 2.51 (2.42–2.59) | <0.01 | 2.51 (2.43–2.60) | <0.01 |
| Urology | 69099 | 1.24 | 10370 | 0.19 | 6.65 (6.14–7.20) | <0.01 | 6.68 (6.22–7.17) | <0.01 |
| Internal Medicine | 349446 | 6.27 | 225426 | 4.06 | 1.55 (1.51–1.59) | <0.01 | 1.55 (1.51–1.58) | <0.01 |
| GP | 130204 | 2.34 | 85330 | 1.54 | 1.52 (1.45–1.60) | <0.01 | 1.53 (1.46–1.60) | <0.01 |
| Chinese Medicine | 205928 | 3.70 | 103709 | 1.87 | 1.98 (1.91–2.05) | <0.01 | 2.01 (1.94–2.08) | <0.01 |
| Family Medicine | 202042 | 3.63 | 138134 | 2.49 | 1.46 (1.41–1.51) | <0.01 | 1.46 (1.41–1.51) | <0.01 |
| Ophthalmology | 84669 | 1.52 | 53206 | 0.96 | 1.59 (1.53–1.65) | <0.01 | 1.59 (1.54–1.64) | <0.01 |
| ENT | 105845 | 1.90 | 57501 | 1.03 | 1.84 (1.77–1.90) | <0.01 | 1.83 (1.76–1.90) | <0.01 |
| Surgery | 57563 | 1.03 | 33353 | 0.60 | 1.72 (1.63–1.82) | <0.01 | 1.73 (1.65–1.81) | <0.01 |
| Psychiatry | 42312 | 0.76 | 10370 | 0.19 | 2.70 (2.46–2.97) | <0.01 | 2.75 (2.50–3.01) | <0.01 |
| ER | 18068 | 0.32 | 8918 | 0.16 | 2.02 (1.90–2.15) | <0.01 | 2.00 (1.89–2.11) | <0.01 |
| Others | 348780 | 6.26 | 220986 | 3.98 | 1.57 (1.54–1.61) | <0.01 | 1.58 (1.55–1.61) | <0.01 |
| **Total** | **1544857** | 27.74 | **942189** | 16.96 | **1.64 (1.61–1.66)** | **<0.01** | **1.64 (1.62–1.66)** | **<0.01** |

*per one person-year. The total-person years were 55697.9 in the IC/BPS group (female: 43935.1) and 55563.5 in the non-IC/BPS group (female: 43810.5).

** IRR: Incidence rate ratio of the IC/BPS group vs. the non-IC/BPS group

***adjusted by age, gender, income, area, hypertension(HTN), diabetes mellitus (DM), hyperlipidemia, and coronary artery disease (CAD).

#only women were included in gynecology. GP: General practitioner; ENT: Ear, nose and throat; ER: Emergency room.

Among the three age groups in both the IC/BPS and non-IC/BPS groups, the rate of OPD visits increased with advanced age in urology. The IRRs of OPD visits in urology were the highest in the younger (less than 40 years) group (18.20; 95% CI, 14.8–22.3), followed by the middle-aged (40–60 years) group (8.04; 95% CI, 7.03–9.20), and the older (over 60 years of age) group (4.42; 95% CI, 4.00–4.85). The increasing trend was also found for other departments, except gynecology. In the gynecology group, the highest IRR was found in the older group (2.81; 95% CI, 2.55–3.10), and the lowest in the younger group (2.42; 95% CI, 2.29–2.55). The detailed data are shown in Table 3.

As for hospitalization, rates were higher in gynecology (45.39 in the IC/BPS group vs. 21.30 in the non-IC/BPS group) (number per 1000 person-year) and urology (34.22 vs. 5.80), with an adjusted IRR of 2.14 (95% CI, 1.83–2.49) and 5.95 (95% CI, 5.01–7.08), respectively. Gender difference was observed in urology hospitalization among IC/BPS patients with an IRR of 23.85 (females) and 72.94 (males), and the adjusted IRR (females to males) was 0.34 (95% CI, 0.29–0.40). The total hospitalization rates (except for gynecology and urology) for the IC/BPS and non-IC/BPS groups were 249.37 vs. 148.66 (number per 1000 person years), with an adjusted IRR of 1.69 (95% CI, 1.58–1.80). The adjusted IRRs were also significantly higher in psychiatry (2.66; 95% CI, 1.65–4.29), colorectal surgery (1.90; 95% CI, 1.32–2.73), internal medicine (1.77; 95% CI, 1.63–1.91), neurosurgery (1.68; 95% CI, 1.39–2.04), and others. The details are listed in Table 4.

The IRR of hospitalizations between the IC/BPS and non-IC/BPS groups was also grouped into three categories by age (Table 5). The finding is similar to OPD visits. Significant higher IRR in younger age comparing to older age subgroup in urology department (IRR: 11.33 v.s. 5.55 and 5.38, p-value of age groups was p < 0.0001). This result is similar in most departments, including psychiatry, orthopedics, neurology, and neurosurgery. The IRR of hospitalization in gynecology is higher IRR in the oldest group (less than 40 years: 1.45 v.s. 40 to 60 years: 2.29 v.s. older than 60 years; 5.70, p-value of age group was p < 0.0001).

**Table 3. Incidence rate ratio (IRR) of outpatient visits between IC/BPS and non-IC/BPS individuals categorized by 3 age groups.**

| Grouping | Age < 40 | | | | | 40≤Age<60 | | | | | Age≥60 | | | | |
|---|---|---|---|---|---|---|---|---|---|---|---|---|---|---|---|
| | IC/BPS (N = 8633) | | non-IC/BPS (N = 8633) | | Adjusted IRR** (95% CI) | IC/BPS (N = 11689) | | Non-IC/BPS (N = 11689) | | Adjusted IRR** (95% CI) | IC/BPS (N = 7668) | | Non-IC/BPS (N = 7668) | | Adjusted IRR** (95% CI) |
| Department | Visits | Rate* | Visits | Rate* | | Visits | Rate* | Visits | Rate* | | Visits | Rate* | Visits | Rate* | |
| Gynecology# | 67981 | 5.14 | 27981 | 2.12 | 2.42† (2.29–2.55) | 70189 | 3.60 | 27860 | 1.43 | 2.52† (2.40–2.64) | 18495 | 1.65 | 6513 | 0.59 | 2.81† (2.55–3.10) |
| Urology | 15522 | 0.90 | 860 | 0.05 | 18.20† (14.8–22.3) | 25459 | 1.09 | 3181 | 0.14 | 8.04† (7.03–9.20) | 28118 | 1.86 | 6329 | 0.42 | 4.42† (4.00–4.89) |
| Internal Medicine | 45176 | 2.62 | 20225 | 1.17 | 2.26† (2.12–2.40) | 132354 | 5.67 | 79718 | 3.42 | 1.67† (1.61–1.73) | 171916 | 11.38 | 125483 | 8.37 | 1.36† (1.31–1.40) |
| GP | 24554 | 1.42 | 12788 | 0.74 | 1.91† (1.76–2.08) | 51335 | 2.20 | 32897 | 1.41 | 1.57† (1.47–1.68) | 54315 | 3.59 | 125483 | 8.37 | 1.37† (1.26–1.48) |
| Chinese Medicine | 67198 | 3.89 | 30507 | 1.77 | 2.21† (2.07–2.35) | 96337 | 4.13 | 47802 | 2.05 | 2.05† (1.95–2.16) | 42393 | 2.81 | 39645 | 2.64 | 1.68† (1.55–1.82) |
| Family Medicine | 33304 | 1.93 | 16904 | 0.98 | 1.97† (1.83–2.13) | 77219 | 3.31 | 53828 | 2.31 | 1.44† (1.37–1.52) | 91519 | 6.06 | 67402 | 4.49 | 1.35† (1.28–1.42) |
| Ophthalmology | 12657 | 0.73 | 6647 | 0.39 | 1.92† (1.79–2.06) | 28126 | 1.21 | 15624 | 0.67 | 1.81† (1.71–1.93) | 43886 | 2.90 | 30935 | 2.06 | 1.41† (1.34–1.48) |
| ENT | 35175 | 2.04 | 18726 | 1.08 | 1.88† (1.77–1.99) | 48972 | 2.10 | 24687 | 1.06 | 1.97† (1.86–2.08) | 21698 | 1.44 | 14088 | 0.94 | 1.52† (1.39–1.66) |
| Surgery | 10454 | 0.61 | 5238 | 0.30 | 2.00† (1.82–2.19) | 24049 | 1.03 | 13558 | 0.58 | 1.78† (1.66–1.91) | 23060 | 1.53 | 14557 | 0.97 | 1.58† (1.44–1.73) |
| Psychiatry | 8663 | 0.50 | 3551 | 0.21 | 3.62† (3.00–4.37) | 13866 | 0.59 | 6999 | 0.30 | 2.82† (2.45–3.25) | 10534 | 0.70 | 5076 | 0.34 | 2.07† (1.77–2.42) |
| ER | 6059 | 0.35 | 2062 | 0.12 | 2.91† (2.59–3.26) | 5468 | 0.23 | 2792 | 0.12 | 1.91† (1.77–2.06) | 6541 | 0.43 | 4064 | 0.27 | 1.58† (1.43–1.74) |
| Others | 79055 | 4.58 | 46078 | 2.67 | 1.72† (1.66–1.79) | 135756 | 5.82 | 86647 | 3.72 | 1.57† (1.52–1.62) | 133969 | 8.87 | 88261 | 5.89 | 1.51† 1.44–1.57) |
| Total | 326309 | 18.91 | 162726 | 9.43 | 2.01† (1.95–2.07) | 618717 | 26.52 | 364552 | 15.64 | 1.71† (1.67–1.75) | 599831 | 39.70 | 414911 | 27.67 | 1.44† (1.40–1.47) |

*per one person-year. The total person-years for the IC/BPS vs. the non-IC/BPS group were as follows: Age<40 group 17256.5 vs. 17260.7; 40≤age<60 group 23332.7 vs. 23307.2; age≥60 group 15108.8 vs. 14995.7.

**Adjusted IRR: Incidence rate ratio, reference group: Non-IC/BPS, adjusted by age, gender, income, area, HTN, DM, hyperlipidemia and CAD.

#only women were included in gynecology. GP: General practitioner; ENT: Ear, nose, and throat; ER: Emergency room

†p < 0.05. The trend test of each pairwise contrast groups was all p<0.05.

## Discussion

This population-based matched-cohort study using the nationwide insurance database indicated that IC/BPS is associated with subsequent increased risk of outpatient visits and hospitalizations. The risks increase not only in the urology and gynecology departments but also in all other clinical departments. Gender difference was observed in urology OPD visits and hospitalizations. The study also showed that males had higher risk compared to females in IC/BPS group. The age effects for IRRs for OPD visits and hospitalizations decreased as age advanced in all departments, except in the gynecology department. Therefore, the impact of IC/BPS on healthcare services involving broader clinical practice and even public policy can be better understood.

Recently, IC/BPS becomes an interesting topic because of not only its large influence on clinical practice but also its economic burden. Chung et al. reported that IC/BPS patients (350 patients) have significantly higher healthcare-related costs (about 1.5-fold) than 1,750 age-

**Table 4. Incidence rate ratio of hospitalizations between IC/BPS and non-IC/BPS individuals.**

| Department | IC/BPS (N = 27990) | | Non-IC/BPS (N = 27990) | | Unadjusted | | Adjusted | |
|---|---|---|---|---|---|---|---|---|
| | No. | Rate* | No. | Rate* | IRR** (95% CI) | p-value | IRR*** (95% CI) | p-value |
| Gynecology# | 1994 | 45.39 | 933 | 21.30 | 2.13 (1.86–2.45) | <0.01 | 2.14 (1.83–2.49) | <0.01 |
| Urology | 1906 | 34.22 | 322 | 5.80 | 5.91 (4.93–7.08) | <0.01 | 5.95 (5.01–7.08) | <0.01 |
| Internal Medicine | 7564 | 135.80 | 4291 | 77.23 | 1.76 (1.61–1.92) | <0.01 | 1.77 (1.63–1.91) | <0.01 |
| Surgery | 1758 | 31.56 | 1146 | 20.63 | 1.53 (1.34–1.74) | <0.01 | 1.54 (1.35–1.76) | <0.01 |
| Colorectal | 266 | 4.78 | 143 | 2.57 | 1.86 (1.23–2.79) | <0.01 | 1.90 (1.32–2.73) | <0.01 |
| Cardiovascular | 88 | 1.58 | 83 | 1.49 | 1.06 (0.73–1.54) | 0.77 | 1.05 (0.73–1.50) | 0.80 |
| Thoracic | 92 | 1.65 | 56 | 1.01 | 1.64 (0.85–3.18) | 0.14 | 1.73 (0.96–3.10) | 0.07 |
| Gastrointestinal | 66 | 1.18 | 51 | 0.92 | 1.29 (0.82–2.03) | 0.27 | 1.36 (0.87–2.11) | 0.18 |
| Other surgeries | 1246 | 22.37 | 813 | 14.63 | 1.53 (1.32–1.78) | <0.01 | 1.53(1.32–1.78) | <0.01 |
| Psychiatry | 1107 | 19.88 | 433 | 7.79 | 2.55 (1.48–4.39) | <0.01 | 2.66 (1.65–4.29) | <0.01 |
| Orthopedics | 1191 | 21.38 | 929 | 16.72 | 1.28 (1.15–1.42) | <0.01 | 1.28 (1.15–1.43) | <0.01 |
| Neurology | 607 | 10.90 | 399 | 7.18 | 1.52 (1.28–1.80) | <0.01 | 1.53 (1.31–1.78) | <0.01 |
| Neurosurgery | 469 | 8.42 | 281 | 5.06 | 1.67 (1.35–2.05) | <0.01 | 1.68 (1.39–2.04) | <0.01 |
| Others | 1160 | 20.83 | 781 | 14.06 | 1.48 (1.30–1.69) | <0.01 | 1.48 (1.30–1.69) | <0.01 |
| Total | 13856 | 249.37 | 8260 | 148.66 | 1.67 (1.56–1.79) | <0.01 | 1.69 (1.58–1.80) | <0.01 |

*per 1,000 person-year; The total person years were as follows: 55697.92 in the IC group, 55563.53 in the non-IC group.

**IRR: Incidence rate ratio of the IC group vs. non-IC group.

***adjusted by age, gender, income, area, HTN, DM, hyperlipidemia, and CAD. #only women were included in gynecology.

matched controls [11]. Payne et al. reported a twofold increase of hospital OPD visits and a threefold increase in the rate of physician office visits related to IC/BPS between 1992 and 2001 from healthcare system in the United States [8]. The annual costs for IC/BPS increased 1.5-fold. Therefore, although IC/BPS accounts for a small percent of healthcare visits, its economic burden is substantial worldwide. Moreover, the true burden of IC/BPS on the healthcare system is probably even higher due to underdiagnosis. The prevalence of IC/BPS is variable to different definitions [12]. IC/BPS is frequently underdiagnosed due to its wide spectrum of symptoms, and these symptoms are easy to be mistaken with other differential diagnosis, like urinary tract infection, overactive bladder, endometriosis, or prostate-related pain [13]. Our study and other reports may remind physicians to pay attention to patients with IC/ BPS in a broader view, in medical, surgical, and mental illness. Adequate recognition of associated or comorbid and possible recommendation or referral for IC/BPS patients can provide not only better healthcare quality but also decrease catastrophic consequences.

It is reasonable that IC/BPS patients have increased risks of OPD visits at the gynecology and urology departments because of typical presentations of IC/BPS as LUTS. After IC/PBS was diagnosed, they will certainly receive treatment and/or follow-ups. Moreover, IC/BPS may be accompanied by other gynecologic symptoms or diseases, e.g., dysmenorrhea [14], endometriosis [15], vulvodynia [16], etc. Tirlapur et al. have reported the prevalence of IC/BPS and endometriosis to be 61% and 70%, respectively, and coexisting IC/BPS and endometriosis was reported to be as high as 48% (16%–78%) among women suffering from chronic pelvic pain in a systemic review (nine studies, 1,016 patients) [17]. The "evil twin syndrome," which means the coexistence of IC/BPS and endometriosis, is one of the difficult-managed syndromes. This makes IC/BPS patients visit the gynecologic OPD even more frequently. Similarly, Ueda et al. indicated that IC/BPS is a common comorbidity among patients with refractory chronic prostatitis/chronic pelvic pain syndrome [18]. Additionally, a recent study of ours showed that IC/ BPS patients are at risk of developing urinary tract cancer at later years [9].

Table 5. Incidence rate ratio of hospitalizations between IC/BPS and non-IC/BPS individuals by three age groups.

| Grouping | Age < 40 | | | | | 40≤Age<60 | | | | | Age≥60 | | | | |
|---|---|---|---|---|---|---|---|---|---|---|---|---|---|---|---|
| | IC/BPS (N = 8633) | | Non-IC/BPS (N = 8633) | | Adjusted IRR** (95% CI) | IC/BPS (N = 11689) | | Non-IC/BPS (N = 11689) | | Adjusted IRR** (95% CI) | IC/BPS (N = 7668) | | Non-IC/BPS (N = 7668) | | Adjusted IRR** (95% CI) |
| Department | Visits | Rate* | Visits | Rate* | | Visits | Rate* | Visits | Rate* | | Visits | Rate* | Visits | Rate* | |
| Gynecology# | 909 | 68.74 | 629 | 47.56 | 1.45* (1.29–1.63) | 768 | 39.34 | 249 | 12.77 | 2.29* (1.61–3.25) | 317 | 28.33 | 55 | 4.96 | 5.70* (2.66–12.22) |
| Urology | 312 | 18.08 | 28 | 1.62 | 11.33* (6.84–18.78) | 681 | 29.19 | 123 | 5.28 | 5.55* (4.17–7.39) | 913 | 60.43 | 171 | 11.40 | 5.38* (4.20–6.90) |
| Internal medicine | 627 | 36.33 | 262 | 15.18 | 2.43* (1.83–3.23) | 1622 | 69.52 | 1000 | 42.91 | 1.69* (1.42–2.00) | 5315 | 351.78 | 3029 | 201.99 | 1.73* (1.56–1.92) |
| Surgery | 291 | 16.86 | 181 | 10.49 | 1.61* (1.19–2.19) | 701 | 30.04 | 433 | 18.58 | 1.67* (1.34–2.06) | 766 | 50.70 | 532 | 35.48 | 1.42* (1.18–1.72) |
| Psychiatry | 525 | 30.42 | 161 | 9.33 | 3.50* (1.83–6.72) | 428 | 18.34 | 187 | 8.02 | 2.54 (1.33–4.85) | 154 | 10.19 | 85 | 5.67 | 1.79 (0.59–5.47) |
| Orthopedics | 155 | 8.98 | 102 | 5.91 | 1.54 (1.14–2.08) | 365 | 15.64 | 278 | 11.93 | 1.35* (1.12–1.62) | 671 | 44.41 | 549 | 36.61 | 1.21* (1.05–1.39) |
| Neurology | 56 | 3.25 | 17 | 0.98 | 3.39* (1.82–6.31) | 124 | 5.31 | 70 | 3.00 | 1.85* (1.30–2.64) | 427 | 28.26 | 312 | 20.81 | 1.36* (1.12–1.64) |
| Neurosurgery | 73 | 4.23 | 27 | 1.56 | 2.73* (1.61–4.64) | 136 | 5.83 | 88 | 3.78 | 1.58* (1.13–2.21) | 260 | 17.21 | 166 | 11.07 | 1.56* (1.19–2.05) |
| Other division | 320 | 18.54 | 180 | 10.43 | 1.78* (1.40–2.26) | 370 | 15.86 | 256 | 10.98 | 1.44* (1.14–1.82) | 470 | 31.11 | 345 | 23.01 | 1.34* (1.09–1.66) |
| Total | 2047 | 118.62 | 930 | 53.88 | 2.25* (1.88–2.68) | 3746 | 160.55 | 2312 | 99.20 | 1.68* (1.48–1.91) | 8063 | 533.66 | 5018 | 334.63 | 1.59* (1.47–1.72) |

*per 1000 person-year. The total person-years for the IC/BPS group vs. the non-IC/BPS group were as follows: Age<40 group 17256.46 vs. 17260.65; 40≤age<60 group 23332.71 vs. 23307.17; age≥60 group, 15108.75 vs. 14995.72

#only women were included in gynecology. The trend test of each pairwise contrast groups was all p<0.05.

In addition to gynecology and urology mentioned above, the IC/BPS patients also have higher OPD visits and hospitalizations in other specialties. Among them, the psychiatric department is the highest in both OPD visits (IRR, 2.75) and hospitalizations (IRR, 2.66). Many studies reported the association between IC/BPS and psychiatric disorders. Watkins et al. reported that depression and panic are developed on as high as 36% and 52% in IC/BPS patients [6]. Clemens et al. reported that mental health disorders were identified in 23% of IC/BPS patients, when compared to 3% in controls (Odd ratio, 8.2; p < 0.001) [19]. A previous study found that patients with IC/BPS are at risk of developing anxiety, depression, and insomnia with a hazard ratio of 2.4 (95% CI, 2.2–2.7), 2.4 (95% CI, 2.2–2.6), and 2.1 (95% CI, 1.8–2.4), respectively [20]. The IRR is also higher in the Chinese medicine department and emergency room. IC/BPS patients tend easily to visit Chinese medicine, which may be due to culture influence. Some Chinese alternative therapies, including Chinese medicine and acupuncture, focus on holistic healing. Some patients may feel some improvement after this kind of alternative therapies [21]. As for more emergency room visits, this may be because it is easy, available, convenient, and low co-pay system of clinical approach in Taiwan. IC/BPS patients may have lower threshold for healthcare seeking, so they tend to visit the emergency room directly once they feel uncomfortable.

IC/BPS is characterized by chronic pain, frequency and urgency voiding, in which these symptoms do not occur in isolation [22]. Considerable overlapping symptoms exist among these patients. Meanwhile, IC/BPS patients have higher incidence of nocturia and urinary urgency than usual. Urgency and/or nocturia may predispose IC/BPS patients to easily falling

down accidents, not only in elderly persons but also in middle-aged woman of more than 40 years old [23,24]. Thus, it can explain the higher IRR of orthopedics and neurosurgery hospitalizations.

The impacts of IC/BPS decrease as age advances, i.e.,., age effect, on healthcare seeking service decreases in urology (IRR of 18.20, 8.04, and 4.42 for age <40, 40 to 60, and >60 years, respectively) and other departments, except in gynecology. Link et al. found that the overall prevalence of symptoms suggestive of IC/BPS is increased in middle-aged adults. As age increases, however, more patients suffered from other symptoms causing from enlarged prostate with bladder outlet obstruction, as in non-IC/BPS controls. This data implied that IC/BPS is more bothersome to younger patients [25]. Our previous study also revealed similar results that younger individuals had increased risks of developing mental illness and insomnia following IC/BPS diagnosis [20]. Similar trends happened in most departments both in OPD visits and hospitalizations, except in gynecology. The age effect on IRR is even higher in older IC/BPS women to seek gynecologic examination. As mentioned above, IC/BPS patients have higher prevalence of some relative gynecologic diseases. In addition to a high coexisting bladder pain syndrome and endometriosis in chronic pelvic pain patients of up to 48% [17], the prevalence of pelvic floor dysfunction in IC/BPS patients ranges from 50% to 85% [26]. Cervigna et al. review showed a close relationship between IC/BPS and pelvic floor dysfunction and vulvodynia. Vulvodynia, most often described as burning pain, occurs in the absence of relevant visible findings. This "burning pain" was reported to be as high as 25% in IC/BPS patients, which is suggestive of a neuropathic pain response [27]. Vulvodynia is estimated to be 11.2-fold (CI 95%, 0.6–203) in IC/PBS patients [28]. This can affect women's sexual life, sex painful and even impossible [16,27]. These symptoms happen even more commonly in elderly women.

Several explanations for the higher risks of OPD visits and hospitalizations were presented. Firstly, IC/BPS patients may have higher mental need of medical healthcare seeking. Nickel et al. had characterized and compared psychosocial phenotypes in IC/BPS women, in which IC/BPS women have significant cognitive and psychosocial alterations compared to controls, including catastrophizing and stress, and to a lesser extent social support [29]. It is reasonable as regards the higher emergency room visit noted in our study. Moreover, it is also based on easy feasibility and low co-pay health insurance system in Taiwan. However, this reason is hard to explain with the higher risks of hospitalizations, which were decided by medical conditions and doctors, instead of patients.

Secondly, IC/BPS patients may have more comorbidities, as compared with non-IC/BPS controls. Nickel et al., in a population-based matched-cohort study, showed that IC/BPS is associated with irritable bowel syndrome, fibromyalgia, and chronic fatigue syndrome [30]. Warren et al. further indicated that IC/BPS patients have more risks of coexisting related non-bladder syndromes, as compared with the control group. Nearly 60% of IC/BPS patients have more than one non-bladder syndromes [28]. Keller et al. reported that up to 32 medical comorbidities are associated with IC/BPS patients [5]. All of these comorbidities may increase subsequent OPD visits and hospitalizations. However, these studies indicated only the cross-sectional correlation between IC/BPS with other diseases; our study further offers a temporal relationship for the subsequent risks of OPD visits and hospitalization in the following two years.

Thirdly, IC/BPS may share some common risk factors; or IC/BPS per se is a precursor (risk factor) for some diseases, as compared with non-IC/BPS controls. Warren et al. reported that several antecedent syndromes are diagnosed more in IC/BPS patients, including migraine, allergies, and sicca syndrome [28]. Many studies implied that IC/BPS is a systemic multi-organ disorder, and the cause may originate from organs outside the bladder. It could induce a

systemic inflammation condition and develop into multiple diseases. Thus, IC/BPS may play an important role as a precursor of some systemic diseases [31].

The strength of this study is its large-scale, nationwide, population-based recruitment. The national insurance system is a single-payer system which covers 98% of the population in Taiwan. Many studies have reported the cross-sectional relationship between IC/BPS and other diseases, but this study offers the temporal relationship and age effects on IC/PBS impacts in the clinical practice.

However, this study still has some limitations resulting from the characteristics of the registry of NHIRD database inherent in this study. Firstly, some potential confounders are lacking that could have an impact on the results, including baseline healthy status, habit of alcohol or tobacco use, and education status due to inherent characteristics of claim data. Secondly, the diagnosis of IC/BPS was dependent on the ICD-9 code within the database, which is diagnosed by the physician using the different diagnostic criteria. In this study, only IC/PBS patients with bladder instillation of sodium hyaluronate were recruited, which requires a strict prereview audit system. Third, distinguishing the disease severity and treatment status of IC/BPS is difficult based on this data source. The different clinical status of IC/BPS patients may have different impacts to OPD visits and hospitalizations. Since our IC/BPS cases are only confined to Hunner type and/or glomerulation, whether the difference between symptom-based and cystoscopic-based diagnosis could influence the patient population remains inclusive according to our data setting. This might need case control trial to compare these two diagnostic criteria setting.

In conclusion, IC/PBS patients were found to be associated with subsequent increased risks of outpatient visits and hospitalizations compared to non-IC/PBS controls, not only in the urology and gynecology departments but also in nearly all specialties. The age effects decrease in urology and in most departments, except in gynecology. This study also provides a broader understanding of IC/PBS within multiple and overlapping systems. These findings and explanations emphasized that IC/PBS patients should be paid more attention due to their systemic complex disease and their need for higher medical service. Thus, our findings broaden our understanding about IC/BPS, from organ-centered to multiple system concepts. A better understanding of IC/BPS patients can provide better healthcare quality.

## Author Contributions

**Conceptualization:** Kun-Lin Hsieh, Ming-Ping Wu.

**Data curation:** Chung-Han Ho, Ming-Ping Wu.

**Formal analysis:** Chung-Han Ho.

**Methodology:** Chung-Han Ho.

**Supervision:** Steven Kuan-Hua Huang, Yao-Chi Chuang.

**Writing – original draft:** Kun-Lin Hsieh, Ming-Ping Wu.

**Writing – review & editing:** Kun-Lin Hsieh, Hung-Yen Chin, Tsia-Shu Lo, Cheng-Yu Long, Chung-Han Ho, Yao-Chi Chuang, Ming-Ping Wu.

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
