## [Decision Letter · Decision Letter 0]

17 Mar 2021

PONE-D-21-02197

Interstitial cystitis/ Bladder pain syndrome patients are associated with subsequent increased risks of out-patient visits and hospitalizations: a population-based study

PLOS ONE

Dear Dr. Wu,

Thank you for submitting your manuscript to PLOS ONE. After careful consideration, we feel that it has merit but does not fully meet PLOS ONE’s publication criteria as it currently stands. Therefore, we invite you to submit a revised version of the manuscript that addresses the points raised during the review process.

ACADEMIC EDITOR:

Based on the comments of the reviewers, I ask that you include your changes very carefully in the revision of the manuscript. Especially the discussion should be more focused and better organized. But also: Make it much clearer how your findings can influence (improve) of the quality of care for people with this syndrome. You also write that the "pathophysiology is unknown." In a large number of people (without Hunner's lesion) the syndrome is not very distinct from overactive bladder syndrome. Can it be determined whether it is excluded in some of the people that specific dysfunction plays a role in the syndrome where the patients refer their signs and symptoms to as pain? And whether you have any information whether the pain has been objectively shown to be related to the lower urinary tract, more specifically, filling (stretching) of the bladder? (are these in local (national) guidelines? as is 'exclusion of other causes', in the international guidelines?)

We look forward to receiving your revised manuscript.

Kind regards,

Peter F.W.M. Rosier, M.D. PhD

Academic Editor

PLOS ONE

Journal Requirements:

2) In the ethics statement in the manuscript and in the online submission form, please provide additional information about the patient records/samples used in your retrospective study, including: a) whether all data were fully anonymized before you accessed them; b) the date range (month and year) during which patients' medical records/samples were accessed.

Reviewers' comments:

Reviewer's Responses to Questions

**Comments to the Author**

1. Is the manuscript technically sound, and do the data support the conclusions?

Reviewer #1: Yes

Reviewer #2: Yes

2. Has the statistical analysis been performed appropriately and rigorously? 

Reviewer #1: Yes

Reviewer #2: Yes

3. Have the authors made all data underlying the findings in their manuscript fully available?

Reviewer #1: Yes

Reviewer #2: Yes

4. Is the manuscript presented in an intelligible fashion and written in standard English?

Reviewer #1: Yes

Reviewer #2: No

5. Review Comments to the Author

Reviewer #1: The manuscript is well written overall. While the data is clearly presented, and it is clear from the manuscript that patients with IC/BPS have higher rates of outpatient visits and higher rates of hospitalization, some of the discussion is conjecture, not based on the data.

The manuscript would be stronger if it simply focused on the finding of higher visits/costs/resource utilization, and did not stray into physiology such as "cross organ sensitization" between the bladder and rectum. While this may be true (i believe it is), it is neither related to nor supported by the data presented.

Furthermore, the concept of hunner's ulcer as "ic" and non-hunner's ulcer as BPS is interesting, and likely true. The presence of hunner's ulcer is likely a different phenotype and different disease process, but it has virtually nothing to do with the finding of increased utilization, visits, hospitalizations, and should be omitted from the manuscript.

Reviewer #2: This is a study of the rate of out-patient visits and hospitalization in a group of patients with a diagnosis of IC/BPS compared to age matched controls. Patients with IC/BPS had more hospitalizations

1. The review criteria for getting the hyaluronate includes the presence of glomerulations or Hunner’s lesions under cystoscopy after hydrodistension- this criteria for glomerulations is no longer used and was part of the old 1988 NIH guidelines. Current guidelines only use symptoms (ref 1 in the paper). Can you comment on how this difference in diagnosis could influence the patient population being studied?

2. Was there a gender difference in the increase in Urology visits compared to controls? In any of the other non GYN visits?

3. Top of page 26, you cite reference 19 for increased rate of cancer in patients with IC/BPS. This is the incorrect reference – should be Wu MP, Luo HL, Weng SF, Ho CH, Chancellor MB, Chuang YC Risk of Urinary Tract Carcinoma among Subjects with Bladder Pain Syndrome/Interstitial Cystitis: A Nationwide Population-Based Study. BioMed Research International. 2018:7495081, 2018.

4. 4. Bottom of page 27- you postulate that the patients go to orthopedics and neurosurgery more due to risk of falls. You reference two articles that deal with the association of LUTS and specifically urgency in older women and men. However 73% of your cohort is under the age of 60. You did not break out the age in table 4.

6. PLOS authors have the option to publish the peer review history of their article (what does this mean?). If published, this will include your full peer review and any attached files.

Reviewer #1: No

Reviewer #2: **Yes: **Michel A. Pontari

---

## [Author Response · Author response to Decision Letter 0]

30 Apr 2021

Response to academic editor and reviewers: 

 We would appreciate our deep thanks for the Academic Editor, and Reviewers’ sophisticated reading on our manuscript, and very instructive and important comments and suggestions. Below were our responses in point-to-point pattern. Enclosed please find our response letter, marked version and clean version. We hope our devotion and revision on the manuscript can merit the acceptance and publication in high impact journal, PLOS ONE. 

PONE-D-21-02197

Interstitial cystitis/ Bladder pain syndrome patients are associated with subsequent increased risks of out-patient visits and hospitalizations: a population-based study

ACADEMIC EDITOR:

Q: Based on the comments of the reviewers, I ask that you include your changes very carefully in the revision of the manuscript. Especially the discussion should be more focused and better organized. But also: Make it much clearer how your findings can influence (improve) of the quality of care for people with this syndrome. 

Response: Thank you for your comments and suggestions. We have revised accordingly. We have revised and organized our discussion to focus on our findings and delete some paragraphs which were irrelevant to our study. 

【p.27, line 260-270】 We deleted the paragraph related to colo-rectal surgery, since many of the relationship between bladder condition and relationship about irritable bowel syndrome, etc are still hypothetic, which is out of the scope of our study. Again, we appreciate the important point of view. 

【p.30, line 325-330】【p.27, line 268-270】 We have deleted the paragraph about pathogenesis, e.g. GAG defect and other mechanism, because of irrelevant information, unrelated to our data. We also delete the theory about “ cross sensitization” since our data could not offer evidence to support it. 

Also, to follow your instructive suggestions, we tried to make more clear how this study can influence the quality of care. “Our study and other reports may remind physicians to pay attention to patients with IC/BPS in a broader view, in medical, surgical and mental illness. Adequate recognition and possible recommendation or referral for IC/BPS patients not only provide better healthcare quality, even decrease catastrophic consequences”【p.25, line 222-225】. And we emphasized that our findings broaden our understanding about IC/BPS, from organ-centered to multiple system concepts in conclusion “This study also provides a broader understanding of IC/PBS within multiple and overlapping systems. These findings and explanations are emphasized that patient with IC/PBS should be paid more attention due to systemic complex disease and higher medical service.”【p.32, line 365-369】 

Q: You also write that the "pathophysiology is unknown." In a large number of people (without Hunner's lesion) the syndrome is not very distinct from overactive bladder syndrome. Can it be determined whether it is excluded in some of the people that specific dysfunction plays a role in the syndrome where the patients refer their signs and symptoms to as pain? And whether you have any information whether the pain has been objectively shown to be related to the lower urinary tract, more specifically, filling (stretching) of the bladder? (are these in local (national) guidelines? as is 'exclusion of other causes', in the international guidelines?) 

Response: Thanks for your comments. Since this is a study based on claim data, the inclusion and exclusion criteria of IC/BPS are up to individual physician’s clinical opinions. However, this study included IC/BPS patients who received bladder instillation treatment, which requires a strict pre-review audit system. There are definite rules in payment regulations of National Health Insurance [Ref National Health Insurance]. “Its provision included: the presence of glomerulation or Hunner’s lesions under cystoscopy after hydrodistention test, voiding diary, negative urine cytology and urine analysis “ [p.6 Material and Methods session, Assembly of study participants]. The above guideline is according to the guideline of Taiwan Urology Association, in line with AUA [Ref TUA guideline] [Ref AUA guideline]

Ref: Taiwan National Health Insurance, https://www.nhi.gov.tw/Content_List.aspx?n=197F438368327FE0&topn=5FE8C9FEAE863B46

Ref: TUA guideline, https://online.fliphtml5.com/thpbe/cljw/#p=1

Ref: AUA guideline, https://www.auanet.org/guidelines/interstitial-cystitis-(ic/bps)-guideline

Reviewers' comments:

Reviewer's Responses to Questions

Comments to the Author

1. Is the manuscript technically sound, and do the data support the conclusions?

Reviewer #1: Yes

Reviewer #2: Yes

Response: Thanks. 

2. Has the statistical analysis been performed appropriately and rigorously? 

 Reviewer #1: Yes

Reviewer #2: Yes

Response: Thanks.

3. Have the authors made all data underlying the findings in their manuscript fully available?

Reviewer #1: Yes

Reviewer #2: Yes

Response: Thanks.

4. Is the manuscript presented in an intelligible fashion and written in standard English?

Reviewer #1: Yes

Reviewer #2: No

Response: After the manuscript revision, we have made substantial English editing. 

5. Review Comments to the Author

Reviewer #1: The manuscript is well written overall. While the data is clearly presented, and it is clear from the manuscript that patients with IC/BPS have higher rates of outpatient visits and higher rates of hospitalization, some of the discussion is conjecture, not based on the data.

The manuscript would be stronger if it simply focused on the finding of higher visits/costs/resource utilization, and did not stray into physiology such as "cross organ sensitization" between the bladder and rectum. While this may be true (i believe it is), it is neither related to nor supported by the data presented. 

Response to Reviewer 1: We agreed the Reviewer’s comment that some discussion opinions were conjecture, not based on the data. We have revised and organized our discussion to more focus on our findings. Including: 

【p.27, line 260-270】 We deleted the paragraph related to colo-rectal surgery, since many of the relationship between bladder condition and relationship about irritable bowel syndrome, etc are still hypothetic, which is out of the scope of our study. Again, we appreciate the important point of view. 

【p.30, line 325-330】【p.27, line 268-270】 We have deleted the paragraph about pathogenesis, e.g. GAG defect and other mechanism, because of irrelevant information, unrelated to our data. We also delete the theory about “ cross sensitization” since our data could not offer evidence to support it.

And we also did other revisions. 

Besides, we tried to pay more attention to influence of the findings. In 2nd paragraph of discussion 【p.24-25】”Our study and other reports may remind physicians to pay attention to patients with IC/BPS in a broader view, in medical, surgical and mental illness. Adequate recognition and possible recommendation or referral for IC/BPS patients not only provide better healthcare quality, even decrease catastrophic consequences. 

Q: Furthermore, the concept of hunner's ulcer as "ic" and non-hunner's ulcer as BPS is interesting, and likely true. The presence of hunner's ulcer is likely a different phenotype and different disease process, but it has virtually nothing to do with the finding of increased utilization, visits, hospitalizations, and should be omitted from the manuscript.

Response: Thanks for your suggestions and comments. In this study, we did not try to discuss on the issue of Hunner-type, or non-Hunner-type, because data inavailability. There are different diagnostic criteria and classifications in IC/BPS. As our response to editor mentioned above, our claim data was collected from NHIRD with inherent claim data characteristics and limitations. We only recruited IC/BPS patients who received bladder instillation treatment after strict pre-review audit system. Therefore our IC/BPS cases only confined to Hunner type (or with glomerulation) due to claim data characteristics. Whether our findings can be deductive to non-Hunner type IC/BPS remains inclusive. It is out of the scope of our study. 

Reviewer #2: This is a study of the rate of out-patient visits and hospitalization in a group of patients with a diagnosis of IC/BPS compared to age matched controls. Patients with IC/BPS had more hospitalizations

Q1. The review criteria for getting the hyaluronate includes the presence of glomerulations or Hunner’s lesions under cystoscopy after hydrodistension- this criteria for glomerulations is no longer used and was part of the old 1988 NIH guidelines. Current guidelines only use symptoms (ref 1 in the paper). Can you comment on how this difference in diagnosis could influence the patient population being studied?

Response: We agreed with the Reviewer’s comment that the finding of glomerulations is no longer used in modern clinical practice. However, due to claim data characteristics, we only recruited IC/BPS patients who received bladder instillation treatment, which requires a strict pre-review audit system. “Therefore our IC/BPS cases only confined to Hunner type (or with glomerulation). Whether this difference between symptom-based and cystoscopic-based diagnosis could influence the patient population remain inclusive according to our data setting. This might need case control trial to compare these two diagnostic criteria setting. It is also out of the scope of our study.”【p. 32, line 356-360】Again, we thank for reviewer’s important and instructive comments, we revised it in the limitation.

Q2. Was there a gender difference in the increase in Urology visits compared to controls? In any of the other non GYN visits?

Response: Yes, we found higher risk of male IC/BPS patients than female in Urology visits and also hospitalization. The rate of Urology visit is female (1.07) and male (2.33) (per one person-year) with adjusted IRR were 0.42 (95% CI: 0.40-0.45). The rate of Urology admission is female (23.85) and male (72.94)(no. per 1000 person-year) with adjusted IRR were 0.34 (95% CI: 0.29-0.40).We had added this data to p.12 and p18. 

Q3. Top of page 26, you cite reference 19 for increased rate of cancer in patients with IC/BPS. This is the incorrect reference – should be Wu MP, Luo HL, Weng SF, Ho CH, Chancellor MB, Chuang YC Risk of Urinary Tract Carcinoma among Subjects with Bladder Pain Syndrome/Interstitial Cystitis: A Nationwide Population-Based Study. BioMed Research International. 2018:7495081, 2018.

Response: We apologize for the mistake. We have corrected the reference number into our text. We appreciated the Reviewer’s sophisticated reading on the manuscript. 

4. 4. Bottom of page 27- you postulate that the patients go to orthopedics and neurosurgery more due to risk of falls. You reference two articles that deal with the association of LUTS and specifically urgency in older women and men. However 73% of your cohort is under the age of 60. You did not break out the age in table 4.

Response: We appreciate the Reviewer’s pointing our mistake. We agreed that the explanation is hypothetic. We further cited a more related reference to explain the possible relationship. Moon SJ et al indicated that higher falling rate in females aged 40 and older due to urgency and LUTS. Also Kurita N et al. found that higher OAB symptoms were associated to high fall risk from 40 year-old and older. [Ref 24 and 25. revision]. These studies showed that high fall risk not only in elderly but also age older than 40. 

24 Moon SJ, Kim YT, Lee TY, Moon H, Kim MJ, Kim SA, et al. The influence of an overactive bladder on falling: a study of females aged 40 and older in the community. Int Neurourol J. 2011; 15(1): 41–7. 

25 Kurita N, Yamazaki S, Fukumori N, Otoshi K, Otani K, Sekiguchi M, et al. Overactive bladder symptom severity is associated with falls in community-dwelling adults: LOHAS study. BMJ Open. 2013; 3(5):e002413

---

## [Decision Letter · Decision Letter 1]

22 Jun 2021

PONE-D-21-02197R1

Interstitial cystitis/ Bladder pain syndrome patients are associated with subsequent increased risks of out-patient visits and hospitalizations: a population-based study

PLOS ONE

Dear Dr. Wu,

Thank you for submitting your manuscript to PLOS ONE. After careful consideration, we feel that it has merit but does not fully meet PLOS ONE’s publication criteria as it currently stands. Therefore, we invite you to submit a revised version of the manuscript that addresses the points raised during the review process.

ACADEMIC EDITOR:

Can You, with the help of the reviewers comments, make a few changes and additions?

We look forward to receiving your revised manuscript.

Kind regards,

Peter F.W.M. Rosier, M.D. PhD

Academic Editor

PLOS ONE

Journal Requirements:

Reviewers' comments:

Reviewer's Responses to Questions

**Comments to the Author**

1. If the authors have adequately addressed your comments raised in a previous round of review and you feel that this manuscript is now acceptable for publication, you may indicate that here to bypass the “Comments to the Author” section, enter your conflict of interest statement in the “Confidential to Editor” section, and submit your "Accept" recommendation.

Reviewer #2: All comments have been addressed

Reviewer #3: (No Response)

2. Is the manuscript technically sound, and do the data support the conclusions?

Reviewer #2: Yes

Reviewer #3: Yes

3. Has the statistical analysis been performed appropriately and rigorously? 

Reviewer #2: Yes

Reviewer #3: Yes

4. Have the authors made all data underlying the findings in their manuscript fully available?

Reviewer #2: Yes

Reviewer #3: Yes

5. Is the manuscript presented in an intelligible fashion and written in standard English?

Reviewer #2: Yes

Reviewer #3: No

6. Review Comments to the Author

Reviewer #2: (No Response)

Reviewer #3: Overall this manuscript is strong given the large data set available for analysis and using a matched case: control study sampling method. The selection of IC/BPS cases on ICD9 code in addition to requiring bladder instillation also allows for strong case selection. A few comments that should be addressed:

1) Methods Section, Lines 104-109. IC/BPS cases are followed from “index date of first registration”, not date of diagnosis. It is unclear how long patients are followed prior to date of initial diagnosis, nor how the inclusion of total time followed impacts the analysis. Or if I am mis-reading this, is index date is equivalent to date of diagnosis? Please add more clarity to describe this.

2) Methods, Lines 128-134: In the analysis of impact of gender or age differences, were these calculated as interaction terms in the model?

3) Methods, Lines 131-133: Please indicate how missing data on covariates was handled. Was missing data coded, or were cases excluded with missing data?

4) Table 1: p-values cannot be 1.0. Recommend change to p>0.9999 for the notation.

5) Results: Lines 152-153: Regarding the gender difference in urology rates, please clarify if these are the overall gender rates for Urology. Or are these the specific for just the IC/BPS group? Clarify the description of the IRR rate that this is female to male “...adjusted IRR rate for females to males was 0.42...”

6) Results, Lines 165-166: Regarding the differences observed across ages, if these rates are increasing, please provide the results of the statistical tests with the appropriate p-value for trend. Were each pairwise contrast of Group 1 to 2 and Group 2 to 3 significantly different?

7) Results, Lines 194-199: Please provide p-values to support these statement of significant differences between age groups.

8) The manuscript would benefit from a detailed editing by a native English speaker.

7. PLOS authors have the option to publish the peer review history of their article (what does this mean?). If published, this will include your full peer review and any attached files.

Reviewer #2: No

Reviewer #3: No

---

## [Author Response · Author response to Decision Letter 1]

29 Jul 2021

We would appreciate our deep thanks for the Academic Editor, and Reviewers’ sophisticated reading on our manuscript, and very instructive and important comments and suggestions. Below were our responses in point-to-point pattern. Also we revised out manuscript under reviewers’ suggestions to improve this paper. Enclosed please find our response letter, marked version and clean version. We hope our devotion and revision on the manuscript can merit the acceptance and publication in high impact journal, PLOS ONE.

Thanks for reviewer 2 and here are responses to reviewer 3:

1) Methods Section, Lines 104-109. IC/BPS cases are followed from “index date of first registration”, not date of diagnosis. It is unclear how long patients are followed prior to date of initial diagnosis, nor how the inclusion of total time followed impacts the analysis. Or if I am mis-reading this, is index date is equivalent to date of diagnosis? Please add more clarity to describe this.

Ans: Thanks for your question. We apologize for the lack of clarity. We have tried to define and revise the sentence. The index date is first registration with the diagnostic code of IC/BPS (ICD 9 code 595.1). Later on, the cases who received bladder instillation as mentioned and at least 3 outpatient visits with the same diagnosis at OPD visit were included. .

We add more information in Lines 110-112.

2) Methods, Lines 128-134: In the analysis of impact of gender or age differences, were these calculated as interaction terms in the model?

Ans: Thanks for your comment. Due to the age and gender had been matched, the interaction terms of age and gender did not be included in the model for avoiding the over adjustments. 

3) Methods, Lines 131-133: Please indicate how missing data on covariates was handled. Was missing data coded, or were cases excluded with missing data?

Ans: Thanks for your question. We apologize for this mistyped error. We want to tell “Some other potential confounders were NOT available due to claim data characters, such as education…”. We correct it in Line 132. Thanks for your careful reading again. 

4) Table 1: p-values cannot be 1.0. Recommend change to p>0.9999 for the notation.

Ans: Thanks for reviewer’s suggestion. The p-value had been modified, accordingly. (We correct in Table 1) 

5) Results: Lines 152-153: Regarding the gender difference in urology rates, please clarify if these are the overall gender rates for Urology. Or are these the specific for just the IC/BPS group? Clarify the description of the IRR rate that this is female to male “...adjusted IRR rate for females to males was 0.42...”

Ans: Thanks for review’s question and suggestion. Yes, our description was not so clear. The gender difference in urology rates is specific for patients with IC/PBS. And the adjust IRR is female to male. We will make more clear description of gender difference in Lines 153-155. 

6) Results, Lines 165-166: Regarding the differences observed across ages, if these rates are increasing, please provide the results of the statistical tests with the appropriate p-value for trend. Were each pairwise contrast of Group 1 to 2 and Group 2 to 3 significantly different?

Ans: Thanks for the reviewer’s comment. The trend test of outpatient visits rates for patients with IC/PBS following the increased age groups had presented as the below table. The trend test of each pairwise contrast groups was also presented. These p-values was added in the below of Table3 (Line 182) 

p-value for trend Overall 40≦ vs. 40-60 40-60 vs. ≧60 40≦ vs. ≧60

Patients with IC/PBS 

Outpatient visits <.0001 <.0001 <.0001 <.0001

7) Results, Lines 194-199: Please provide p-values to support these statement of significant differences between age groups.

Ans: Thanks for the reviewer’s comment. The differences between age groups all show p-value<.0001. These p-values were added in the manuscript to explain the statement of differences between age groups. The paragraph was modified as below. (Line 200-207)

“Significant higher IRR in younger age comparing to older age group in urology department (IRR: 11.33 vs. 5.55 and 5.38, p-value of age groups was p<.0001). This result is similar in most departments, including psychiatry,

orthopedics, neurology and neurosurgery. The IRR of hospitalization in gynecology is higher IRR in oldest group (less than 40 years: 1.45 vs. 40 to 60 years: 2.29 vs. older than 60 years: 5.70, p-value of age groups was p<.0001). ” 

8) The manuscript would benefit from a detailed editing by a native English speaker.

Ans: Thanks for review’s suggestion. Due to the authors are not native English speakers, we sent the manuscript to international editing company for help. This new version of manuscript had been edited by the editors from Enago.

---

## [Decision Letter · Decision Letter 2]

17 Aug 2021

Interstitial cystitis/bladder pain syndrome patient is associated with subsequent increased risks of outpatient visits and hospitalizations: a population-based study

PONE-D-21-02197R2

Dear Dr. Wu,

We’re pleased to inform you that your manuscript has been judged scientifically suitable for publication and will be formally accepted for publication once it meets all outstanding technical requirements.

Kind regards,

Peter F.W.M. Rosier, M.D. PhD

Academic Editor

PLOS ONE

Additional Editor Comments (optional):

Reviewers' comments:

Reviewer's Responses to Questions

**Comments to the Author**

1. If the authors have adequately addressed your comments raised in a previous round of review and you feel that this manuscript is now acceptable for publication, you may indicate that here to bypass the “Comments to the Author” section, enter your conflict of interest statement in the “Confidential to Editor” section, and submit your "Accept" recommendation.

Reviewer #2: All comments have been addressed

Reviewer #3: All comments have been addressed

2. Is the manuscript technically sound, and do the data support the conclusions?

Reviewer #2: Yes

Reviewer #3: Yes

3. Has the statistical analysis been performed appropriately and rigorously? 

Reviewer #2: Yes

Reviewer #3: Yes

4. Have the authors made all data underlying the findings in their manuscript fully available?

Reviewer #2: Yes

Reviewer #3: Yes

5. Is the manuscript presented in an intelligible fashion and written in standard English?

Reviewer #2: Yes

Reviewer #3: Yes

6. Review Comments to the Author

Reviewer #2: (No Response)

Reviewer #3: Very well presented findings. The authors have addressed all concerns raised and the editing has improved the manuscript substantially.

7. PLOS authors have the option to publish the peer review history of their article (what does this mean?). If published, this will include your full peer review and any attached files.

Reviewer #2: No

Reviewer #3: No

---

## [Editor Report · Acceptance letter]

27 Aug 2021

PONE-D-21-02197R2 

Interstitial cystitis/bladder pain syndrome patient is associated with subsequent increased risks of outpatient visits and hospitalizations: a population-based study 

Dear Dr. Wu:

I'm pleased to inform you that your manuscript has been deemed suitable for publication in PLOS ONE. Congratulations! Your manuscript is now with our production department. 

Kind regards, 

on behalf of

Dr. Peter F.W.M. Rosier 

Academic Editor

PLOS ONE